



# Diffusivity measurements of volatile organics in levitated viscous aerosol particles

Sandra Bastelberger[1], Ulrich K. Krieger[1], Beiping Luo[1], and Thomas Peter[1]

[1]Institute for Atmospheric and Climate Science, ETH Zürich, 8092 Zürich, Switzerland

*Correspondence to:* S. Bastelberger (sandra.bastelberger@env.ethz.ch)

**Abstract.** Field measurements indicating that atmospheric secondary aerosol (SOA) particles can be present in a highly viscous, glassy state have spurred numerous studies addressing low diffusivities of water in glassy aerosols. The focus of these studies is on kinetic limitations of hygroscopic growth and the plasticizing effect of water. In contrast, much less is known about diffusion limitations of organic molecules and oxidants in viscous matrices. These may affect atmospheric chemistry

and gas-particle partitioning of complex mixtures with constituents of different volatility. In this study, we quantify the diffusivity of a volatile organic in a viscous matrix. Evaporation of single particles generated from an aqueous solution of sucrose and small amounts of volatile tetraethylene glycol (PEG-4) is investigated in an electrodynamic balance at controlled humidity (RH) and temperature. The evaporative loss of PEG-4 as determined by Mie resonance spectroscopy is used in conjunction with a radially resolved diffusion model to retrieve translational diffusion coefficients of PEG-4. Comparison of the experimentally

derived diffusivities with viscosity estimates for the ternary system reveals a breakdown of the Stokes-Einstein relationship, which has often been invoked to infer diffusivity from viscosity. The evaporation of PEG-4 shows pronounced RH and temperature dependencies and is severely depressed for RH $\lesssim$ 30 %, corresponding to diffusivities $< 10^{-14}$ cm$^2$ s$^{-1}$ at temperatures $< 15$ °C. The temperature dependence is strong, suggesting a diffusion activation energy of about 300 kJ mol$^{-1}$. We conclude that atmospheric volatile organic compounds can be subject to severe diffusion limitations in viscous organic aerosol particles.

This may enable an important long-range transport mechanism for organic material, including pollutant molecules such as polycyclic aromatic hydrocarbons (PAHs).

## 1 Introduction

Volatile organic compounds (VOCs) emitted into the atmosphere can undergo gas-phase oxidation, lowering the volatility of some reaction products sufficiently to partition into the particle phase and form so-called secondary organic aerosol (SOA).

SOA is exposed to complex heterogeneous and condensed phase aging processes such as oxidation by hydroxyl radicals (OH) and ozonolysis. Through these processes, SOA may form intricate mixtures of potentially highly functionalized constituents with varying volatility and hygroscopicity known to comprise a substantial fraction of semivolatile organic compounds (SVOCs) (Jimenez et al., 2009; Donahue et al., 2014). The finding that atmospheric SOA can take on a highly viscous, semi-solid or even glassy state has drawn attention to its physical state and how low humidity and temperature facilitate

glass formation (Zobrist et al., 2008; Murray, 2008; Virtanen et al., 2010; Koop et al., 2011; Bateman et al., 2015, 2016). As





condensed phase diffusivity is generally expected to be inversely related to viscosity, diffusion coefficients of water, organic molecules and oxidants are essential to understanding the influence of SOA physical state on aerosol processes. Slow diffusivities of organic molecules and oxidants can affect atmospheric chemistry, a point illustrated in a study by Davies and Wilson (2015), who observed the formation of interfacial gradients in the reactive uptake of OH radicals on viscous citric acid (CA)

aerosol particles. Whereas particles remained well-mixed when dilute, they showed that the surface region affected by the reaction (characterized by the depletion of CA and the formation of reaction products) is a strong function of RH below 50 %, implying condensed phase diffusion limitations. Li et al. (2015) studied ammonia reactive uptake on SOA and determined threshold RH values at which reactivity is not limited by condensed phase diffusion spanning a large range from < 5 % for isoprene derived SOA to > 90 % RH for $\beta$-caryophyllene derived SOA. Using coated-wall flow tube experiments Steimer

et al. (2015) investigated the reaction kinetics of the ozonolysis of shikimic acid. They showed how ozonolysis rates increase with the uptake of water, providing evidence that water acts as plasticizer and thereby induces changes in the physical state. Studies by Lignell et al. (2014) and Hinks et al. (2016) suggest that low diffusion rates can also influence photochemistry. Modeling studies demonstrate that the timescale of SOA partitioning and size distribution dynamics are, amongst other factors such as volatility, determined by bulk diffusivity (Zaveri et al., 2014). Kinetic gas-particle modeling for the photo-oxidation

of dodecane under dry conditions by Shiraiwa et al. (2013) revealed that experimental particle-size distributions were better represented when taking the finite condensed phase diffusivity into account. Ye et al. (2016) investigated the influence of RH on the mixing of SVOCs and SOA produced from $\alpha$-pinene ozonolysis and toluene photo-oxidation using quantitative single-particle mass spectroscopy and isotopic labeling. While condensed phase diffusion at room temperature was not rate-limiting for equilibration at RH > 40 % for either SOA type, the uptake of semivolatile vapors by toluene-based SOA was impeded at

low RH whereas there was no indication of limitations in $\alpha$-pinene-derived SOA. Similarly, Liu et al. (2016) studied the RH dependence of the evaporative mass loss from films representative of SOA derived from anthropogenic or biogenic sources. Unlike biogenic films, for which no RH threshold was observed, anthropogenic films exhibited increased evaporation rates above RH $\sim$ 20-30 %. Diffusion limitations for oxidants and organic molecules could also lead to the trapping of VOCs, such as harmful polycyclic aromatic hydrocarbons (PAH) in the aerosol phase and protect them from oxidation, thereby facilitating

their long-range transport. In a study of evaporation rates, Abramson et al. (2013) estimated the condensed phase diffusivity ($D_c$) of pyrene, a PAH, contained in freshly produced $\alpha$-pinene SOA as $D_c = 2.5 \times 10^{-17}\,\mathrm{cm^2\,s^{-1}}$. Zhou et al. (2013) coated benzo[$a$]pyrene-ammonium sulfate (BaP-AS) particles with products from $\alpha$-pinene ozonolysis and observed limitations in BaP reaction with ozone for RH below $\sim$50 %, from which they estimated $D_c \sim 5 \times 10^{-14}\,\mathrm{cm^2\,s^{-1}}$.

Clearly, the breadth of processes affected by condensed phase diffusion limitation indicates a need to determine diffusivities in viscous aerosol and their dependence on RH and temperature. Many studies have relied on rheological measurements (e.g. poke-flow, bead mobility (Renbaum-Wolff et al., 2013) and coalescence in optical tweezers (Power et al., 2013)) in combination with the Stokes-Einstein relation to predict diffusivity from viscosities of SOA or SOA model systems. According to





Stokes-Einstein, the diffusivity $D_c$ of a diffusing molecule with hydrodynamic radius $r_H$ is given by

$$D_c = \frac{k_B T}{6\pi\eta r_H} \quad , \tag{1}$$

where $k_B$ is the Boltzmann constant, $T$ is the temperature and $\eta$ is the dynamic viscosity of the medium. However, substantial decoupling of diffusivity and viscosity has been observed in various systems for temperatures close to or below the glass

transition temperature $T_g$ (Corti et al., 2008a, b). Measured water diffusivities may exceed Stokes-Einstein predictions by several orders of magnitude (Power et al., 2013; Price et al., 2014). Several bulk and levitated aerosol particle techniques, based on isotopic tracers, fluorescence and mass transport measurements, have emerged to measure water diffusivity in highly viscous systems (Zobrist et al., 2011; Price et al., 2014; Lienhard et al., 2015; Davies and Wilson, 2016; Marshall et al., 2016). Few studies have addressed the diffusion of larger molecules (Champion et al., 1997; Price et al., 2016; Chenyakin et al., 2017)

such as VOCs. Marshall et al. (2016) studied the suppression of the evaporation of maleic acid in viscous sucrose aerosol and its dependence on RH, but stopped short of ascribing diffusivities, choosing instead to describe diffusion limitations qualitatively as apparent reduction in vapor pressure.

In this study, we present quantitative diffusivity measurements of PEG-4, which is a volatile organic with a well-established vapor pressure representative of atmospheric SVOCs. We investigate the evaporation of PEG-4 from levitated sucrose aerosol

particles, a viscous matrix used as SOA proxy, as function of RH and temperature. The diffusivities will be retrieved from experimental radius data using a radially resolved diffusion model.

## 2   Experiment

We illustrate the main features of our experiments in Fig. 1 in terms of the measured radius evolution (red line) of a ternary sucrose/PEG-4/water particle. Here, three sections (A, B, C) at nearly constant relative humidities characterize the condensed

phase diffusion controlled evaporation of PEG-4. They are separated by two rapid uptakes of water in response to sudden moistening. The steepening of the slopes with increasing humidities reveals the plasticizing effect of water, enabling PEG-4 to leave the droplet faster when RH is higher. If the plasticizing effect of water did not occur, modeling using constant condensed phase diffusivities (blue line) would even predict a slight flattening of the slopes. This shows qualitatively that condensed phase diffusion has to be taken into account for the prediction of evaporative loss in viscous SOA. In the following, we describe the

electrodynamic balance (EDB) set-up, the experimental procedure and in a subsequent chapter the diffusion model which allows us to model the radius evolution seen in Fig. 1 quantitatively.





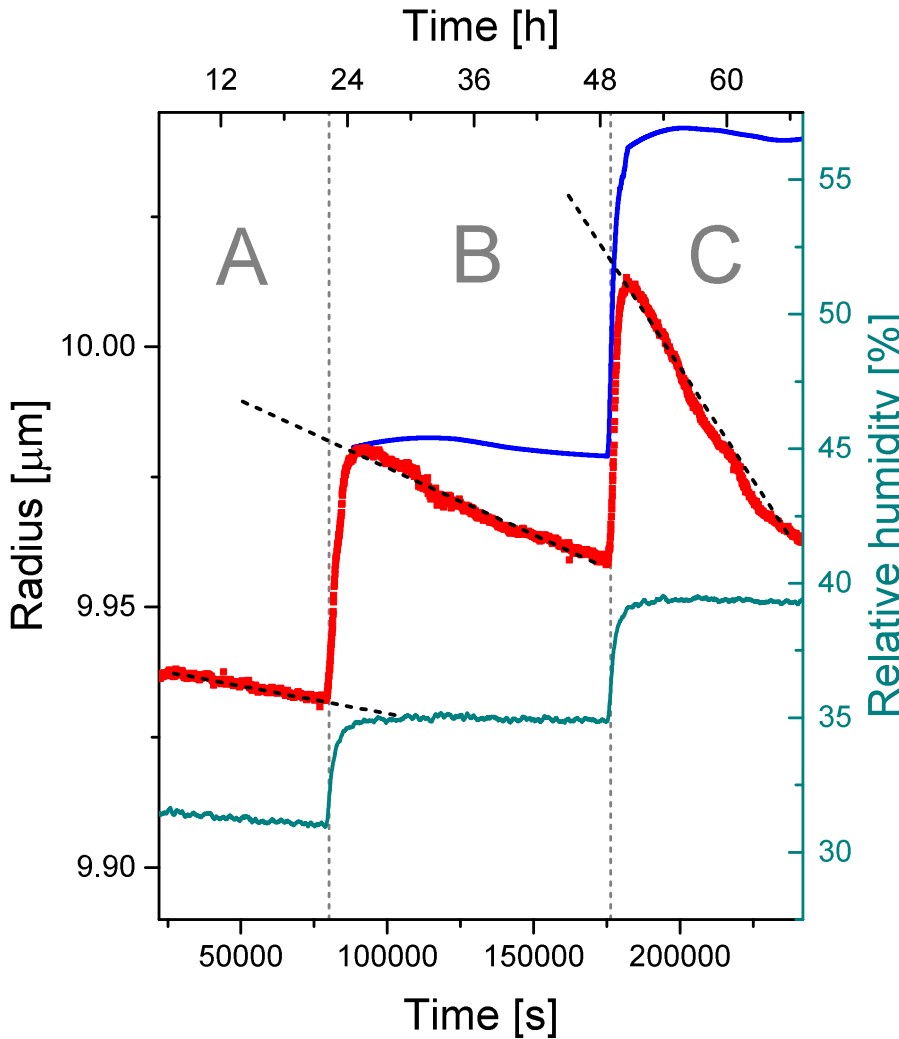

**Figure 1.** Evolution of the measured radius (red line) of an evaporating sucrose/PEG-4/water droplet at 10 °C. The experiment is divided into three sections (A, B, C) during which RH (dark cyan line) was kept constant. The slopes of the radius (straight black dotted lines) steepen in response to the step-increase in RH. A comparison with a hypothetical evolution of the radius under the assumption that the PEG-4 condensed phase diffusivity of section A is applicable in section B and C reveals the plasticizing effect of water.





## 2.1 Set-up

The experiments for this study were conducted in the set-up previously described in-depth by Lienhard et al. (2014) and Krieger et al. (2000). An inductively charged single particle is injected into a double-ring EDB (two plane-parallel AC electrode rings between two DC endcaps) using an ink jet cartridge. The particle is trapped in an electric field generated by the DC and AC

potentials applied to the electrodes. A feedback control based on CCD camera imaging of the particle's position continuously adjusts the DC potential to compensate the gravitational pull or any other vertical forces that may be present, such as drag caused by the gas flow passing through the trap. The EDB is embedded in a triple-wall glass chamber in which well defined pressure, temperature and RH conditions covering the atmospheric range can be established. The trap has an insulation vacuum between the two outer walls and a heat transfer fluid is pumped through the two inner walls of the chamber, stabilizing the

temperature better than 0.1 K with an accuracy of $\pm 0.5$ K. The relative humidity within the chamber can be regulated by adjusting the mixing ratio of a dry and humidified nitrogen gas flow (with a total flow of 20 sccm). A capacitive RH probe positioned in the center of the upper DC electrode provides RH data representative of the immediate gas-phase environment of the particle (distance $\sim 1$ cm).

   Particle composition and size can be characterized based on mass (exploiting the proportionality of DC-potential and mass),

refractive index and radius. In this study, two Mie resonance spectroscopy based methods were used: (i) The particle is illuminated using a tunable diode laser (TDL) scanning a range of 765–781 nm. High resolution Mie resonance spectra (Steimer et al., 2015) are acquired at $90°$ scattering angle for both TE and TM polarization. As the scattering by spherical droplets yields characteristic patterns of resonances, refractive index and radius can be determined simultaneously with an accuracy better than 0.005 in refractive index and a corresponding accuracy in size of $\sim 2 \times 10^{-3}$ µm (ii) The backscatter signal of a

broad-band LED source centered around the sodium-D line (589 nm, $\Delta\lambda \sim 35$ nm) is recorded using a spectrograph (slow scan back-illuminated CCD array detector) which is able to resolve wavelength shifts of the Mie resonance positions associated with changes in size and refractive index (Zardini et al., 2006; Zardini and Krieger, 2009). The radius is calculated from these wavelength data as described in more detail below.

## 2.2   Measurements

The chemical compounds used to prepare the ternary system were as follows: Millipore water (resistivity $> 18.2$ M$\Omega$ cm, total organic content $< 5$ppb), sucrose (Arcos Organics 99.7 %) and tetraethylene glycol (Sigma-Aldrich $> 99$ %). All compounds were used without further purification. Individual aerosol particles were generated at ambient conditions from a diluted aqueous solution ($\simeq 2 - 2.5$ % weight fraction solute) with initial PEG-4 content in terms of mole fraction of total solute $f_{\mathrm{PEG}} =$

$n_{\mathrm{PEG}}/(n_{suc} + n_{\mathrm{PEG}}) \leq 0.15$. Excess water rapidly evaporates, yielding particles with radii between 7 - 12 µm. The chamber pressure was set to 800 hPa. Throughout the run of each experiment, the temperature was set to a constant value between 8 - 19.5 °C. The experiments were conducted as shown in Fig. 2: After injection, particles were rapidly dried and kept at constant low humidity. Subsequently, particles were repeatedly exposed to step increases in RH followed by prolonged periods





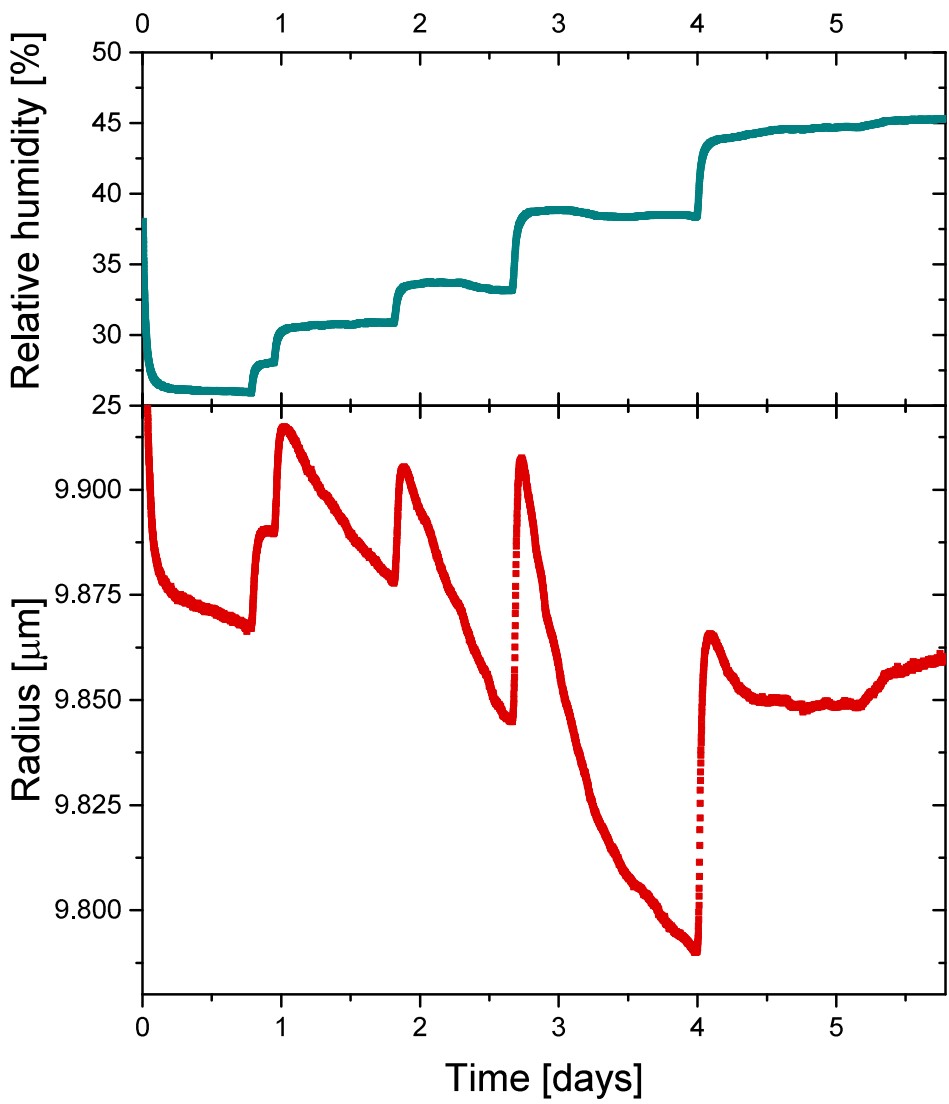

**Figure 2.** Experiment on a sucrose/PEG-4/water particle at 19.5 °C conducted in an electrodynamic balance. Upper panel: Prescribed relative humidity versus time. Lower panel: Radius of particle as determined using LED Mie resonance spectra.

of constant RH (> 10 h), finally attaining RH values > 45 %.

Throughout the experiment, low resolution LED Mie resonance spectra were recorded to monitor the radius and composition change of the particle. After the last RH step, particles were allowed to fully equilibrate with the gas phase until they followed the equilibrium thermodynamics of aqueous sucrose. The corresponding equilibrium radius $r_0$ was precisely determined using





high resolution Mie spectra, thereby also ascertaining the number of moles sucrose contained within the particle; information vital for the conversion of LED raw spectra into radius data. Given the slow evaporation process and long equilibration times, the total duration of each experiment exceeded 4.5 days.

In addition, an EDB measurement on a particle with initial $f_{\mathrm{PEG}} \sim 0.8$ aimed at the determination of the ternary system activity coefficient of PEG-4, $\gamma_{\mathrm{PEG}}$, was performed. To ensure an evaporation not limited by condensed phase diffusion, RH was set to $\sim 74\%$ at $T = 17\,°\mathrm{C}$. As in the measurements described above, LED spectra were recorded throughout the experiment and the equilibrium radius was measured using TDL spectra after equilibration.

## 2.3 Processing of LED spectra

Elastic scattering on dielectric spheres yields a series of characteristic resonances of different orders and modes as function of the Mie parameter $\chi = 2\pi r/\lambda$, with $r$ being the radius of the particle and $\lambda$ the wavelength. The resonance position is determined by the refractive index as described by Mie theory. Change in particle radius and composition (and thereby refractive index) is thus reflected in shifts of the Mie resonance positions in wavelength space. For a given resonance at position $\lambda_0$ associated with $r_0$ and refractive index of binary sucrose $m_0$, the radius $r(t)$ at time $t$ can be inferred from the resonance position $\lambda(t)$ and refractive index $m(t)$ with

$$r(t) = \left[1 - K(m(t), \chi)\left(\frac{m(t) - m_0}{m_0}\right)\right]\frac{\lambda(t) r_0}{\lambda_0}, \tag{2}$$

where $K(m(t), \chi)$ assumes values between 0.94 and 0.95 for $1.30 \leq m \leq 2.00$ and $\chi \sim 100$ (Ray et al., 1991). Here, $K$ was treated as a constant and set to 0.95. As PEG-4 evaporates and RH varies, $m(t)$ will change as implicit function of the PEG-4 and water concentration and can be calculated based on molar refractivities derived from binary solution data and the Lorentz-Lorenz-relation (aqueous sucrose refractive index from Rosenbruch et al. (1975); pure PEG-4 refractive index as given by Sigma-Aldrich, procedure as described in Luo et al. (1996)). The Zdanovskii-Stokes-Robinson approximation was used to calculate concentrations for a given $f_{\mathrm{PEG}}$ and RH (see Sect. 3). As $f_{\mathrm{PEG}}(t)$ constitutes an unknown but for the final part of an experiment when the particle has equilibrated, Eq. (2) has to be solved iteratively tracing back in time starting from $r_0$ and $f_{\mathrm{PEG},0} = 0$ by incrementing $f_{\mathrm{PEG}}(t)$.

## 3 Diffusion model

In general, the concentration changes due to condensed phase diffusion that occur during the evaporation of an aerosol particle are described by Fick's second law. Considering the sphericity of a liquid droplet, we use the following expression in spherical coordinates

$$\frac{\partial c}{\partial t} = \frac{1}{r^2}\frac{\partial}{\partial r}\left(r^2 D_c \frac{\partial c}{\partial r}\right), \tag{3}$$

where $c$ is the molar concentration, $D_c$ denotes the condensed phase diffusivity of the diffusing species, $t$ is the time and $r$ is the distance from the particle center. In atmospheric systems, we expect $D_c$ to depend both on temperature and composition





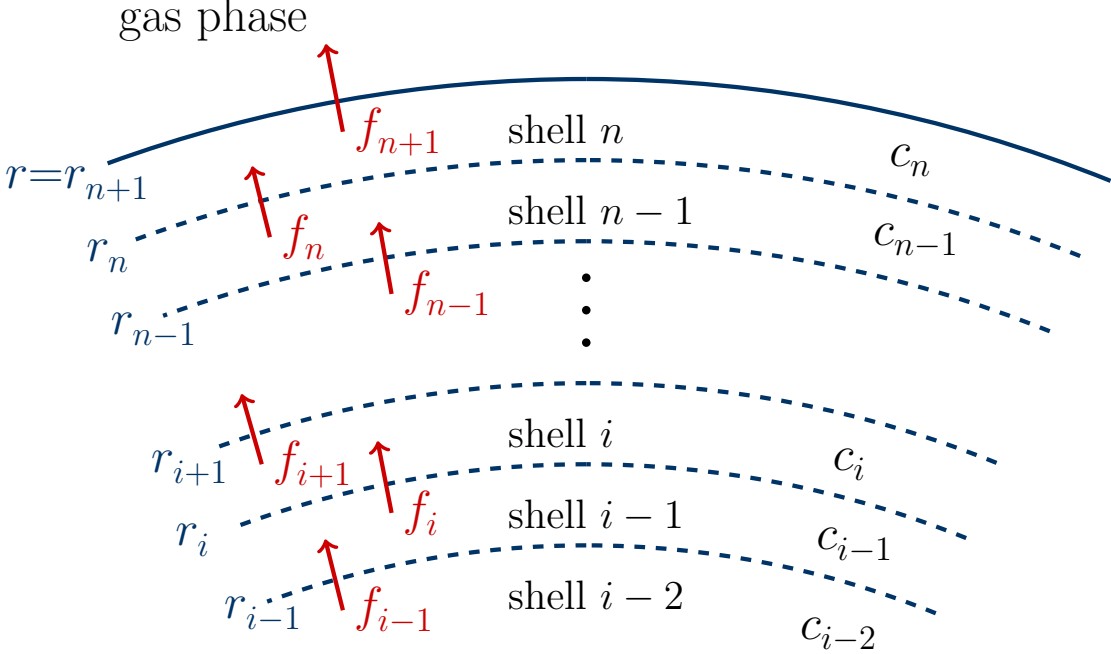

**Figure 3.** Illustration of the discretization scheme used in the diffusion model: A spherical particle of radius $r$ is divided into $n$ concentric shells. An individual shell $i$ has an outer radius $r_{i+1}$ such that $r = r_{n+1}$. In each time step the concentration $c_i$ in shell $i$ is recalculated by balancing the diffusion fluxes $f$ in and out of the shell. The flux into the gas phase $f_{n+1}$ is calculated separately using Eq. (6).

because low viscosity constituents such as water may act as plasticizers. Subsequently, we solve Eq. (3) using a numerical approach.

The diffusion model presented in this study is tailored to our experiments and treats the evaporation and diffusion of a low quantity of PEG-4 in a sucrose-water matrix. It is adapted from the Euler forward step based diffusion model described in
5  Zobrist et al. (2011), which was originally developed to solve the nonlinear form of Eq. (3) for the diffusion of water. While the diffusing species is now PEG-4, our new model follows the same principles. In short, the problem is discretized in time steps as well as spatially by dividing the particle into $n$ concentric shells (see Fig. 3). Each shell $i$ extending from $r_i$ to $r_{i+1}$ contains a certain concentration $c_i$ of the diffusing species PEG-4. In each time step, we calculate the surface integrated molar flux between two adjacent shells $i-1$ and $i$ with

10  $$f_i = 4\pi r^2 D_c \frac{[c_i - c_{i-1}]}{0.5[r_{i+1} - r_{i-1}]}; \qquad \forall i \in \{2, ..., n\}, \tag{4}$$





where $D_c$ is the condensed phase diffusivity and the fraction represents the concentration gradient across the shell interface at $r_i$. The flux into the gas phase $f_{n+1}$, which incorporates gas phase diffusion, is treated separately. Balancing the fluxes yields the change of number of moles $\Delta N_i$ of the diffusing species

$$\Delta N_i = [f_i - f_{i+1}] \Delta t; \qquad \forall i \in \{1, ..., n\}, \tag{5}$$

from which concentrations and corresponding shell radii are recalculated. The time steps $\Delta t$ are chosen such that $\Delta N_i$ does not exceed 1 % for all $i$. The following changes were implemented to adapt the model to the sucrose/PEG-4/water-system:

1. *Water within the particle is in equilibrium with the gas phase at all times.* Under our experimental conditions, water diffusion is fast enough to reach equilibrium on timescales much shorter than the time intervals over which PEG diffusion coefficients were determined. Therefore, it is not treated explicitly. The number of moles sucrose in each shell is kept constant.

2. $D_c = D_{PEG}$ *is not a function of PEG-4 concentration and solely depends on water content and $T$.* While water is a plasticizer and the diffusivity of PEG-4 in the matrix depends on the water content, PEG-4 is assumed not to have a plasticizing effect on the sucrose matrix at the concentrations used in the experiments. Thus, at constant RH, the problem is effectively reduced to solving the linear form of the diffusion equation.

3. *The water content of the ternary system is estimated via the Zdanovskii-Stokes-Robinson approximation.* The molality of the ternary system is derived from the molalities of the binary solutions (Stokes and Robinson, 1966). The water activity of aqueous PEG-4 was parametrized as described in Appendix A1. For aqueous sucrose, the parametrization of Zobrist et al. (2011) was used.

4. *The density of the ternary system is calculated from partial molar volumes.* See Appendix A2.

5. *The gas-particle phase partitioning of PEG-4 is described using the modified Raoult's law.* Assuming zero PEG-4 concentration at infinite distance from the particle, the flux into the gas phase $f_{n+1}$ in the continuum regime reads

$$f_{n+1} = 4\pi r_{n+1} \mathcal{D}_g \frac{x_n \gamma_{\text{PEG}} p^0}{RT}, \tag{6}$$

where $\mathcal{D}_g$ is the gas phase diffusion constant of PEG-4, $x_n$ is the mole fraction of PEG-4 in shell $n$, $\gamma_{\text{PEG}}$ is the activity of PEG-4 in the ternary system (see Appendix A3), $p^0$ is the pure component vapor pressure of PEG-4 and $R$ is the universal gas constant.

While we can rely on well-established methods to estimate gas-phase diffusivities (Bird et al., 2007; Krieger et al., 2017) and precise vapor pressure data is available for PEG-4 (Krieger et al., 2017), the condensed phase diffusivity $D_{\text{PEG}}$ and activity coefficient $\gamma_{\text{PEG}}$ of PEG-4 are unknowns that have to be retrieved from experimental data. As illustrated by Soonsin et al. (2010),



$\gamma_{\mathrm{PEG}}$ can be determined in the gas-diffusion limited regime, where $D_{\mathrm{PEG}}$ is sufficiently high such that PEG-4 is distributed homogeneously throughout the particle, by measuring the radius change of evaporating particles. In contrast, it is impossible to further constrain $D_{\mathrm{PEG}}$ in this regime. $D_{\mathrm{PEG}}$ is only accessible experimentally when condensed phase diffusion limitations apply and the resulting radial concentration profiles are reflected in the evaporation kinetics. As the determination of $D_{\mathrm{PEG}}$ still

requires knowledge of $\gamma_{\mathrm{PEG}}$, we rely on Wilson's approach for multi-component systems (Orye and Prausnitz (1965), Appendix A3) to parametrize $\gamma_{\mathrm{PEG}}$ based on binary and ternary bulk data as well as EDB $\gamma_{\mathrm{PEG}}$ measurements in the gas diffusion limited regime.

## 4 Results and discussion

### 4.1 Activity coefficients

As activity coefficients of organics are generally only weakly temperature dependent (Ganbavale et al., 2015), we assume $\gamma_{\mathrm{PEG}}$ to be constant over the investigated temperature range of 8-19.5 °C. We determine $\gamma_{\mathrm{PEG}}$ at high RH by freely adjusting its value in the diffusion model to reproduce the experimental radius curve presented in Fig. 4a (red) while $D_{\mathrm{PEG}}$ was set to a constant value of $1.0 \times 10^{-9}$ cm$^2$ s$^{-1}$. The chosen diffusivity is well within the gas phase diffusion limited regime and can be justified using simple Stokes-Einstein estimates (which tend to be reliable at high RH). Assuming a hydrodynamic radius

of $r_H = 0.4$ nm = 4 Å for PEG-4 (Kuga (1981); for more information, see section 4.3), the viscosity of binary sucrose at a RH of 74 % (Quintas et al., 2006) implies $D_{\mathrm{PEG}} \simeq 5 \times 10^{-9}$ cm$^2$ s$^{-1}$. For comparison, the experimental diffusivity of sucrose in sucrose at room temperature and the same RH was shown to be $2.0 \times 10^{-9}$ cm$^2$ s$^{-1}$ (Price et al., 2016). Taking the high PEG-4 content and its smaller viscosity into account, the actual diffusivity is likely to be even higher. The modeling results as seen in Fig. 4a do not show large deviations from ideal behavior (i.e. $\gamma_{\mathrm{PEG}} = 1$ ) with $\gamma_{\mathrm{PEG}}$ assuming values between 0.42

to 0.75 in the studied concentration range. The simulation results are robust for $D_{\mathrm{PEG}} > 2.0 \times 10^{-11}$ cm$^2$ s$^{-1}$ as diffusion is sufficiently fast to maintain a near constant PEG-4 concentration within the particle, see Fig. 4b. Careful examination of the whole experimental data set and conservatively taking the diffusivity of sucrose in aqueous sucrose as reference revealed further cases where mass loss was gas phase diffusion limited, that allowed for determination of $\gamma_{\mathrm{PEG}}$. Note that PEG-4 diffusion limitations that go unrecognized will distort $\gamma_{\mathrm{PEG}}$ values derived from evaporation experiments, leading to an underestimate.

The experimentally determined values for $\gamma_{\mathrm{PEG}}$ are summarized in Fig. 5a, see filled circles for measurements under conditions with high diffusivities. They were used to parametrize $\gamma_{\mathrm{PEG}}$ according to Wilson's equation (Fig. 5b, see Appendix A3). Rings with labels refer to hypothetical values, which are distorted by low diffusivities and will be discussed in Sect. 4.2. A comparison with AIOMFAC calculations (Zuend et al., 2008; Topping et al., 2016), though deviating by a factor of 0.9-1.8 from AIOMFAC calculations in the relevant concentration range, show a dependency similar to our Wilson parametrization, supporting our findings (Fig. 5c). The Wilson parametrization of the activity of PEG-4 was implemented into the diffusion

model to calculate the flux of PEG-4 into the gas phase (see Eq. (5)).





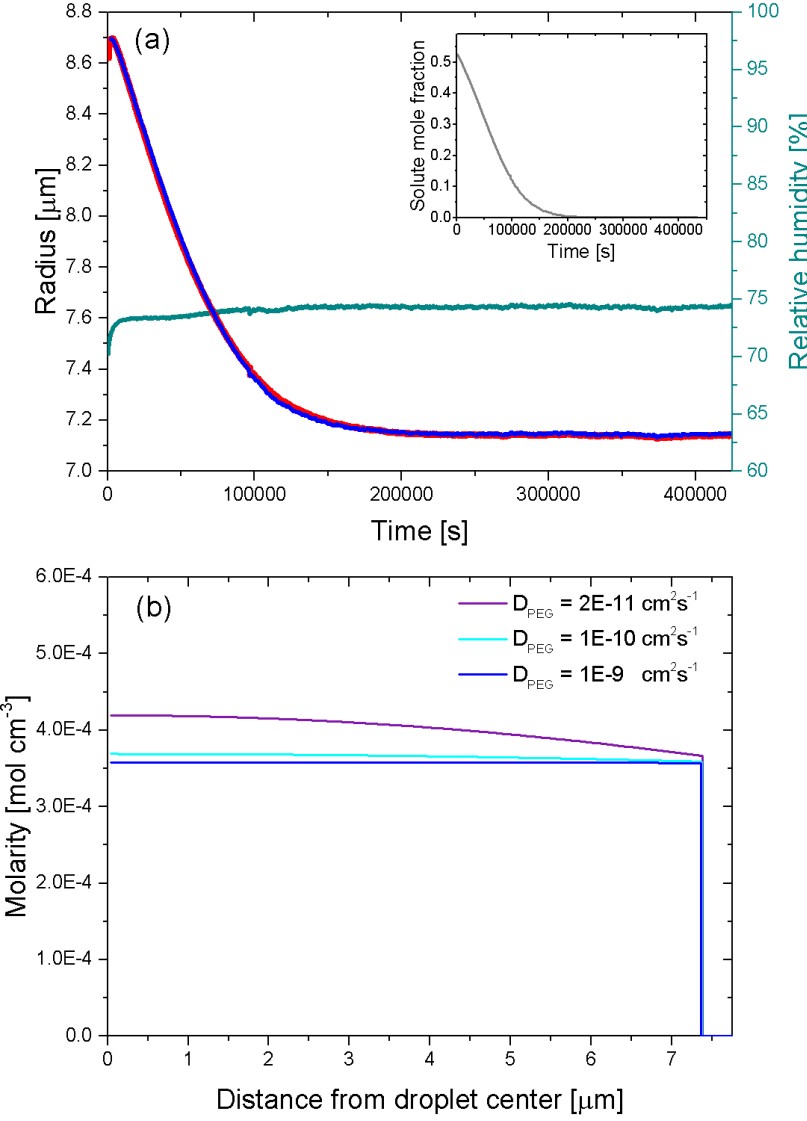

**Figure 4.** Five-day evaporation measurement in a sucrose/PEG-4/water droplet at 17 °C . (a) Prescribed relative humidity (dark cyan, right scale), measured radius (red curve) and modeling result (blue curve) assuming a condensed phase diffusivity of $1.0 \times 10^{-9}$ cm$^2$ s$^{-1}$. Insert: Derived solute mole fraction $f_{\text{PEG}}$ vs. time. (b) Model radius and PEG-4 molarity profile within particle at $t = 100000$ s assuming different diffusivities but using the same activity dependence.





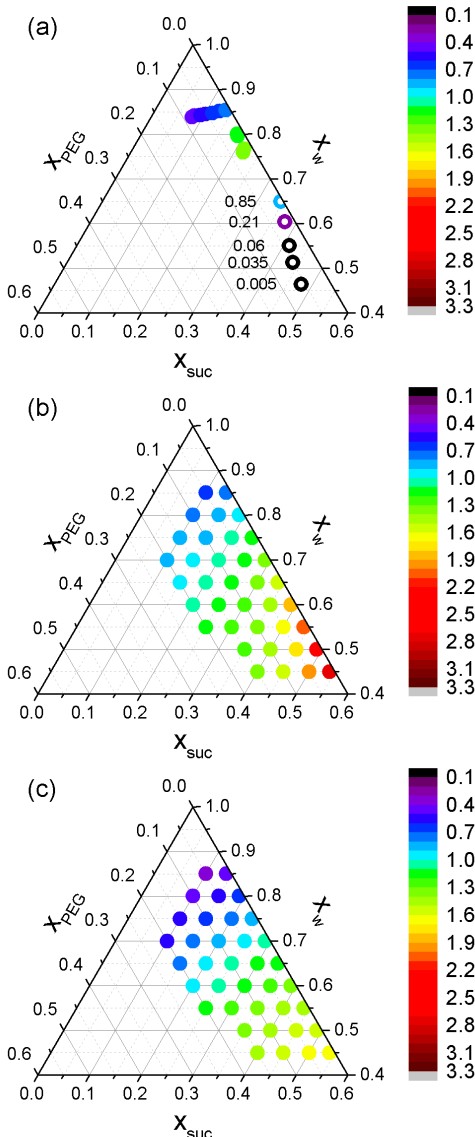

**Figure 5.** Activity coefficient $\gamma_{PEG}$ as function of composition. $x_{suc}$, $x_w$, and $x_{PEG}$ denote the mole fractions of sucrose, water and PEG-4, respectively. Symbol color: Absolute value as indicated by color scale. (a) Filled circles indicate $\gamma_{PEG}$ as determined from evaporation experiments under conditions where condensed phase diffusion is sufficiently fast. Rings with labels represent hypothetical $\gamma_{PEG}$ values needed to explain experimental evaporation curves in Fig. 6 if no diffusion limitations applied (see section 4.2). (b) $\gamma_{PEG}$ as given by Wilson parametrization derived from experiment activity data (circles in (a) and binary solution data). (c) $\gamma_{PEG}$ as determined by AIOMFAC calculations (Zuend et al., 2008; Topping et al., 2016).



## 4.2 Determination of Diffusivities

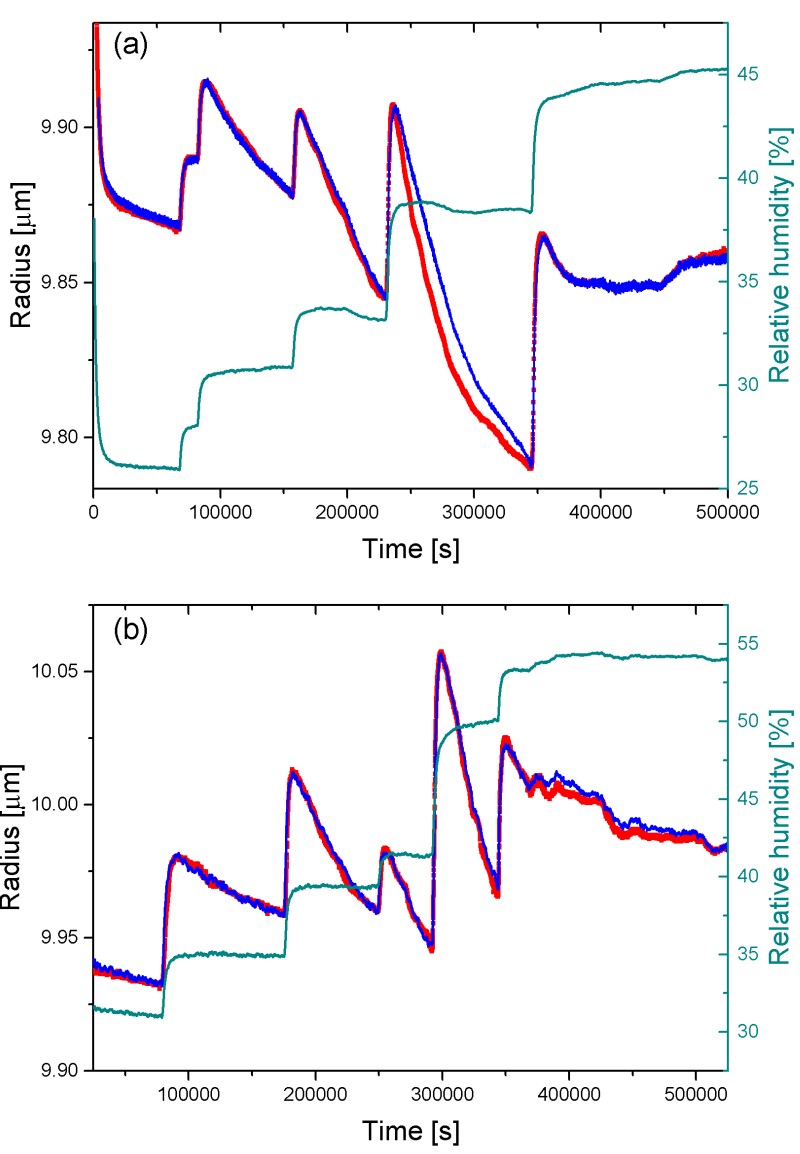

**Figure 6.** Experimentally derived (red) and modeled (blue) radius and measured relative humidity (dark cyan) vs. time. Diffusion coefficients were adjusted to reproduce experimental radius curve. (a) Measurement at 19.5 °C. (b) Measurement at 10.0 °C.





In Fig. 6 we show raw experimental data and modeled radius evolution. Initially the particle shrinks rapidly (red line) as RH in the chamber decreases (dark cyan line). Whereas binary sucrose particles will eventually approach a constant radius if kept at fixed RH, the continuous evaporation of PEG-4 in the ternary system results in a persistent radius change. The subsequent RH step causes a steep radius increase due to the hygroscopic growth of the particle. Note that the rate of the radius

change increases following the humidity steps, contrary to what one would expect from diluting a homogeneous particle. In this case, $x_{\mathrm{PEG}}$, and therefore the vapor pressure, would decrease. If one adheres to the notion of a homogeneous particle for argument's sake (i.e. sufficiently fast condensed phase diffusion), the observed size change can only be explained with an activity coefficient strongly increasing with RH, thereby counteracting the decrease in vapor pressure due to continuous decrease in PEG-4 content. The $\gamma_{\mathrm{PEG}}$ dependence (Fig. 5a, labeled rings) this line of thought requires is neither consistent with

the Wilson parametrization nor the trend seen in AIOMFAC calculations, as seen in Fig. 5b and c. It follows that the decrease in evaporation at low RH can only be explained with a depression in surface concentration of PEG-4 as a direct consequence of condensed phase diffusion limitations. The increase in evaporation ensuing an RH step can consequently be attributed to the plasticizing effect water has on the sucrose matrix: The associated increase in diffusivity facilitates the transport of PEG-4 from the center of the particle to the surface where it can evaporate, leading to higher evaporation rates.

It is evident from the sharp RH steps in Fig. 6 that there is no time delay between the particle response and the model response (blue curve), thereby validating the assumption that water diffusion is sufficiently fast and does not have to be treated explicitly under the experimental conditions considered in this study. The diffusion coefficients $D_{\mathrm{PEG}}$ were assumed to be constant between RH steps and were chosen such that they best replicate the slope of the experimental radius curve at a given RH level. As seen in the case of the final RH step (> 52 %) of Fig. 6b, the evaporation rate in the model cannot attain high enough

values to reproduce the experimental radius curve even when arbitrarily fast diffusivities are chosen. Because evaporation is governed by gas phase diffusion at such high RH, this can be attributed to our activity parametrization slightly underpredicting $\gamma_{\mathrm{PEG}}$ and/or an error in vapor pressure and illustrates the inability of our method to determine condensed phase diffusivities in the gas phase diffusion limited regime. Only data points that are clearly in the condensed phase diffusion limited regime, i.e. the particle exhibits a nonhomogeneous concentration profile, were considered in the final $D_{\mathrm{PEG}}$ data set presented in Fig 7.

The measurements at 19.5 °C (red squares), 15 °C (green triangles) and 10 °C (blue circles) follow a clear temperature and RH dependence, decreasing with decreasing RH and temperature. For comparison, diffusivities of other molecules in aqueous sucrose are shown. The diffusivities of fluorescein, rhodamine 6G and calcein (Chenyakin et al., 2017) in aqueous sucrose at 21.4 °C are roughly an order of magnitude smaller than the PEG-4 diffusivities at 19.5 °C. While PEG-4 is smaller and lighter, the molecular masses as well as hydrodynamic radii of these fluorescent dyes are comparable or larger than sucrose

and measured diffusivities seem to agree with the diffusivity of sucrose in aqueous sucrose measured by (Price et al., 2016) at 23.5 °C. As expected, the PEG-4 diffusivities fall below the diffusivity of water in aqueous sucrose (Zobrist et al., 2011).

Due to the complexity of the model it is unwieldy to perform classical error propagation. Errors will arise from uncertainties in composition, density, refractive index, RH, $\gamma_{\mathrm{PEG}}$ and pure component vapor pressure of PEG-4 (as discussed for Fig.

6b), and the human factor in judging what diffusion coefficients best represent experimental size change data. The error in RH





was estimated to be $\pm 1.5$ % above and $\pm 3$ % below 30 % RH. To address the error in pure component vapor pressure and $\gamma_{\mathrm{PEG}}$, the experimental data was reanalyzed with the flux given by Eq. (5) multiplied or divided by a factor of 3. This factor was chosen as a conservative estimate as the Wilson $\gamma_{\mathrm{PEG}}$ generally agrees well with AIOMFAC predictions and measured activities. Further, the error in pure component vapor pressure is expected to be less than $\pm$ 10 %. While applying this factor

5 had no discernable effects for $D_{\mathrm{PEG}} < 10^{-13}$ cm$^2$ s$^{-1}$, the uncertainty in vapor pressure and $\gamma_{\mathrm{PEG}}$ strongly affects the retrieved $D_{\mathrm{PEG}}$ values at high humidities in proximity to the gas-phase diffusion limited regime as reflected in the large error for the data point measured at the highest RH. The error for lower diffusivities was judged to be at the most a factor of 1.4 of the absolute diffusivity value.



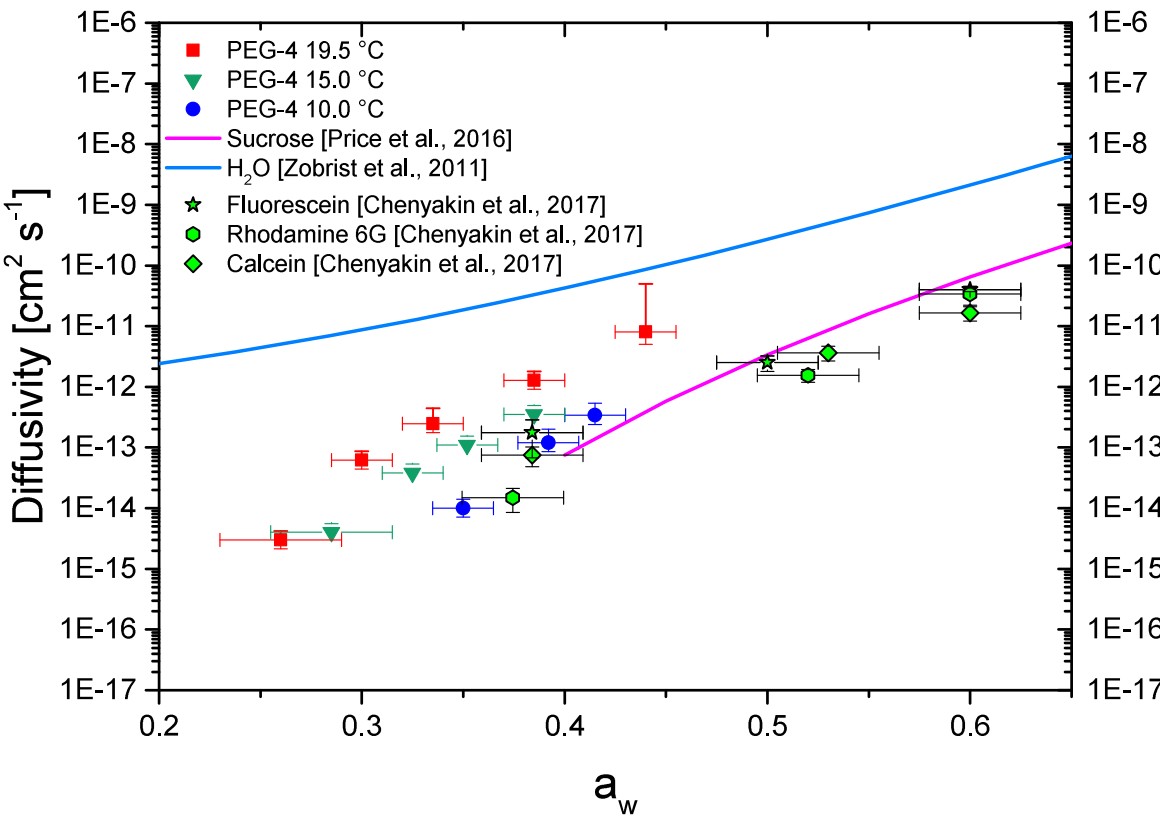

**Figure 7.** Experimentally determined diffusivity of PEG-4 vs. water activity and comparison with diffusivities of other compounds in aqueous sucrose. This study: PEG-4 diffusivity at 19.5 °C (red squares), 15 °C (green triangles) and 10 °C (blue circles). Light green symbols with black outline represent diffusivities of the fluorescent dyes fluorescein (stars), rhodamine 6G (hexagons) and calcein (diamonds) in aqueous sucrose as determined by Chenyakin et al. (2017). The light blue curve is a parametrization of the diffusivity of water in aqueous sucrose (Zobrist et al., 2011) at 19.5 °C. The pink curve is a parametrization of the diffusivity of sucrose in aqueous sucrose (Price et al., 2016) at 23.5 °C.





### 4.3 Stokes-Einstein Comparison

In the following, the validity of the Stokes-Einstein relationship will be assessed. Unfortunately, whilst viscosities of sucrose/water mixtures were measured over a large RH range, no viscosity data is available for our ternary system. Marshall et al. (2016) highlighted that adding a third chemical compound to a binary matrix may significantly influence its viscosity by comparing the viscosities of a water/sucrose/maleic acid system with that of aqueous sucrose at a given RH for higher maleic acid content. But given the generally low values of $f_{\text{PEG}}$ in this study, the viscosities of the ternary system should not significantly deviate from the viscosity of the binary aqueous sucrose at the same RH. On the contrary, a strong dependence of diffusivity on PEG-4 content would imply that the diffusivity needed to model the radius change between two RH steps strongly decreases over time as PEG-4 evaporates, but modeling indicates that diffusivities at constant RH do not change. Thus a strong effect at the PEG-4 levels considered here is not substantiated by experimental data.

Fig. 8 shows a comparison of the measured PEG-4 diffusivities with diffusivities calculated from literature viscosity data for aqueous sucrose (Song et al., 2016; Quintas et al., 2006). To estimate the hydrodynamic radius, data for PEG 200, i.e. a polydisperse mixture of polyethylene glycols with an average molecular weight of $200 \, \text{g mol}^{-1}$ comparable to PEG-4 with $194.23 \, \text{g mol}^{-1}$, were considered. Values given in the literature vary between $r_H$ = 3.4 and 5.2 Å (Dohmen et al., 2008), but the effect of this range of possible $r_H$ on the Stokes-Einstein estimate is negligible compared to the uncertainty caused by viscosity. For this study, $r_H$ was assumed to be 4 Å (Kuga, 1981). To correct for a possible bias caused by a plasticizing effect of PEG-4, rather than comparing data points of the same RH, the Stokes-Einstein comparison presented here is sucrose mole fraction $x_{suc}$ based. This is motivated by a recent study of Song et al. (2016) who investigated the viscosity of mixtures and tested amongst other approaches the validity of a mixing rule presented by Bosse (2005), which is mole-fraction based and uses pure component viscosities to predict viscosities of binary mixtures. Bosse ideal mixing yields good results for aqueous sucrose and suggests that, if applicable to our ternary system, sucrose provides the dominant contribution to the mixture viscosity due to its very high "pure component" viscosity and the contribution of PEG-4 can be treated as water in first approximation. The room temperature RH-based viscosity parametrization presented by Song et al. (2016) (which includes the Power et al. (2013) data corresponding to water activities as low as 0.3) was plotted as function of $x_{suc}$ using the Zobrist et al. (2011) $a_w$ parametrization (Stokes-Einstein derived diffusivities given as purple curve). In comparison to our 19.5 °C measurement, Stokes-Einstein underestimates $D_{\text{PEG}}$ about a factor $\sim$ 140-600 over the whole experimental range. The break-down of Stokes-Einstein seems much more pronounced if the measured diffusivities are compared to the temperature dependent Williams-Landel-Ferry (WLF) parametrization by Quintas et al. (2006) ($\sim$ 3-4 orders of magnitude). Note that, in contrast to the Song et al. (2016) parametrization, we extrapolate far beyond the scope of the experimental data the parametrization is based on, see blue (10 °C) and red (20 °C) hollow symbols in Fig. 8. Because of the exponential concentration dependence the magnitude of the deviation between our data and the parametrization is questionable. Yet, it might provide qualitative insight into the temperature dependence. Though more pronounced in our data, the temperature dependence of the diffusivity is clearly present




in the Stokes-Einstein estimates. In addition, the magnitude of the temperature dependence increases with sucrose concentration in both our measurements and the Stokes-Einstein predictions based on the Quintas et al. (2006) parametrization.

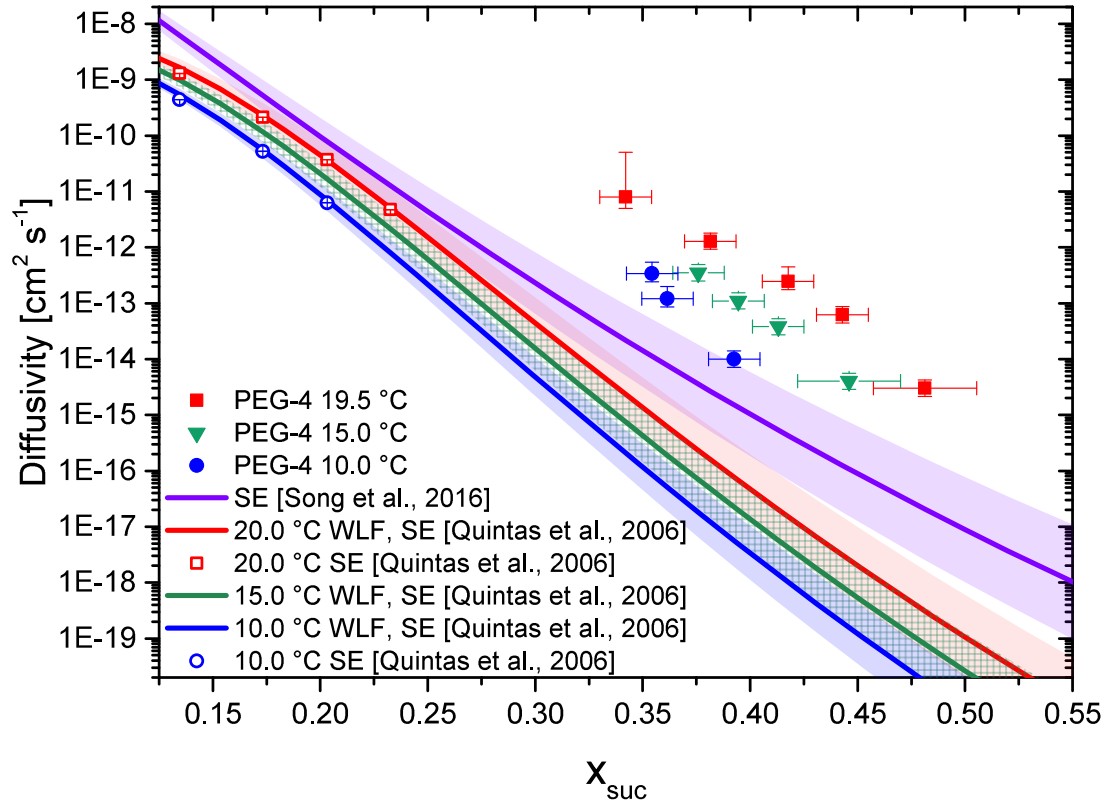

**Figure 8.** Comparison of experimental data with Stokes-Einstein estimates vs. mole fraction sucrose $x_{suc}$. Measured PEG-4 diffusivities (filled symbols) in the ternary system at 19.5 °C (red squares), 15 °C (green triangles) and 10 °C (blue circles). Stokes-Einstein diffusivity estimates (curves) using $r_H$ = 4 Å (shaded regions indicate error envelopes pertaining to the uncertainties in measured viscosities) were calculated for the room temperature viscosity parametrization of Song et al. (2016) (purple) and the WLF viscosity parametrization of Quintas et al. (2006) (red 20 °C, green 15 °C and blue curve 10 °C). Equally, Stokes-Einstein diffusivity estimates were calculated for measured viscosities from Quintas et al. (2006) (hollow symbols, red 20 °C, blue 10 °C).

The high-viscosity breakdown of Stokes-Einstein reported in this study and by others (Power et al., 2013; Price et al., 2015; Chenyakin et al., 2017) begs the question of how the hydrodynamic radii relate to the magnitude of the discrepancy

5    between prediction and measured diffusivities. We try to address this question in Fig. 9, which shows reported diffusivities of





various molecules (Water, PEG-4, sucrose, fluorescein, rhodamine 6G and calcein) in a sucrose matrix plotted against their hydrodynamic radius. The paucity of data renders an accurate comparison of data points corresponding to the same sucrose-mole fraction difficult, but all points in the figure correspond to $x_{suc}$ between 0.38 and 0.41 and temperatures close to room temperature and should therefore allow for a comparison. In addition, reported values of the hydrodynamic radius vary greatly

5   as evident from the figure. Therefore, we chose to only provide a range of radii. It is obvious that measured diffusivities for the smallest hydrodynamic radii are orders of magnitude larger than the Stokes-Einstein prediction. Further, they appear to decrease much stronger with increasing hydrodynamic radius than suggested by the $1/r$ dependence of the Stokes-Einstein relation (purple curve). Instead, the data seem to suggest an exponential dependence of the diffusivity on the hydrodynamic radius for radii up to $\sim 8$ Å.




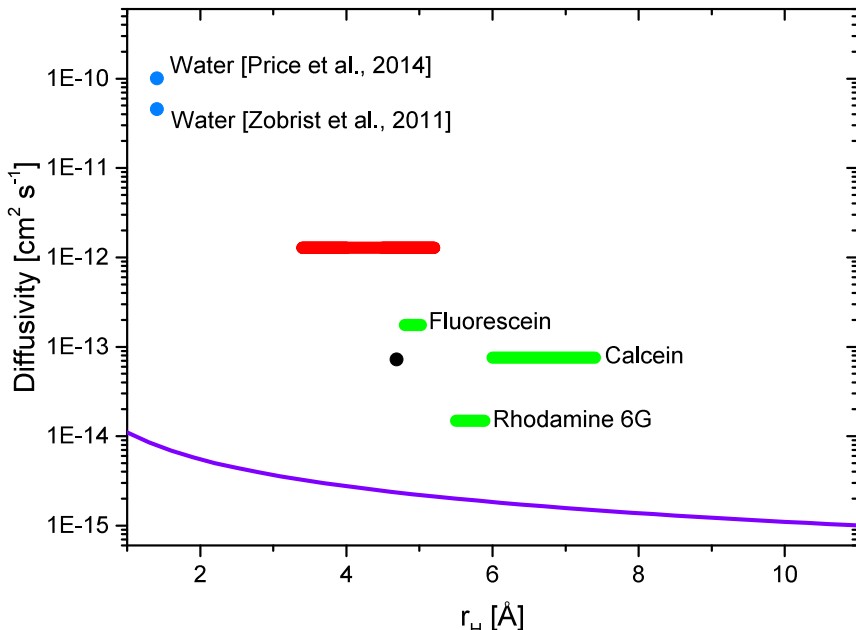

**Figure 9.** Comparison between experimental diffusivities and Stokes-Einstein prediction for molecules of various hydrodynamic radii $r_H$ at room temperature. The diffusivity of water is shown in light blue and was taken from the parametrizations of Zobrist et al. (2011) and Price et al. (2014). The data points for fluorescein, rhodamine 6G and calcein, shown in green, were taken from Chenyakin et al. (2017). PEG-4 (this study, 19 °C) is shown red. The black data point represents sucrose as calculated from the parametrization of Price et al. (2014). All data points correspond to values of $x_{suc}$ between 0.38 and 0.41. Hydrodynamic radius ranges were taken from the literature (Robinson and Stokes (2012); Mustafa et al. (1993); Ribeiro et al. (2006); Zarnitsyn et al. (2008); Müller et al. (2008); Tamba et al. (2010); Majer and Zick (2015); table 4 from Dohmen et al. (2008)). The Stokes-Einstein estimates (purple line) were calculated using the Song et al. (2016) viscosity parametrization for $x_{suc} = 0.381$ as function of $r_H$.



## 5  Atmospheric outlook

Diffusion limitations for oxidants and organic molecules could disable or at least decelerate aerosol heterogeneous chemical reactions. Further, they could disable or reduce the aerosol-induced transport of certain species, which cannot be taken up by particles after becoming highly viscous, or, conversely, enable or enhance the aerosol-induced transport of species, which were

locked in before becoming highly viscous. Zelenyuk et al. (2012) suggested that long-range transport of polycyclic aromatic hydrocarbons (PAHs) trapped in highly viscous SOA may contribute to unexpectedly high particle-bound PAH concentrations in remote regions. In addition, there have been suggestions that PAHs may be trapped irreversibly within the soot matrix (Fernández et al., 2002), but we do not further discuss this mechanism here.

The evaluation of organic pollutants in regional air quality models reveals that models underestimate PAH concentrations (va-

por and particulate) by a factor of 4 in comparison with measurement stations in the cold winter season, whereas there is good agreement in summer (Efstathiou et al., 2016). Furthermore, model-measurement discrepancies are larger for Arctic sites than for mid-latitudes (Friedman et al., 2014). Despite uncertainties in PAH emissions, which have their own seasonal and latitudinal dependencies (e.g., Harrison et al. (1996); Abdel-Shafy and Mansour (2016)) that need to be taken into account, these results suggest that that models underestimate the role of temperature in long-range transport of PAHs.

Friedman et al. (2014) tested a number of long-range transport scenarios in a global chemical transport model, such as (i) PAHs and SOA undergoing instantaneous reversible gas-particle equilibrium or (ii) PAHs being trapped in SOA or (iii) in aerosols consisting of primary organic matter and black carbon. They find (i) to be incapable to reproduce observed PAH concentrations in the mid-latitudes and the Arctic, whereas (ii) improves the modeling substantially and (iii) provides an even better fit. The model relies on simplifying assumptions for both (ii) and (iii): PAH evaporation from the aerosols is described by a $1/e$-

equilibration time $\tau = 4.3$ d, and the combined effect of PAH evaporation and oxidation in the condensed phase is described by $\tau = 1.4$ d. These times $\tau$, based on Zelenyuk et al. (2012), are in the same ballpark as $\tau$'s derived from our measurements for room temperature, and the best agreement with (iii) found by Friedman et al. (2014) is probably due to the strong correlation between the emission of primary organic matter and black carbon and the emission of PAHs. However, dependences of these simple process parameterizations on particle radius $r$ or on condensed phase diffusivities $D_c(x_w, T)$, which is a strong function

of $T$, are neglected. As we show below, the temperature dependence of $D_c$ implied by our measurements will result in much longer equilibration times at ambient temperatures below room temperature.

The validity of long-range transport mechanisms for semi-volatile pollutants may hinge on the magnitude of the increase in evaporation times associated with slow condensed phase diffusivity and its temperature dependence. Relying on the thermodynamic and kinetic properties of the ternary sucrose/PEG-4/water system characterized in this work, we explore a scenario

in which an aerosol particle with radius $r = 100$ nm containing a small amount of a volatile organic species has been rapidly transported upwards through the troposphere, where it is mixed into a cold ambient air with RH = 38 % that is void of the dissolved volatile compound (i.e. PEG-4 partial pressure is zero). The diffusion model introduced in Sect. 3 was used to determine the equilibration time $\tau$, i.e. the time after which the amount of the volatile species (here PEG-4) within the particle has dropped to $1/e$ of its initial value. The initial mole fraction is assumed to be $f_{\text{PEG}} = 0.04$. Temperatures have been varied from




10 to -40 °C (but keeping relative humidity constant at 38 %).

In passing we note that the aerosol particle is in the transition regime, requiring a correction factor relative to a continuum regime treatment. Its magnitude is about $\sim 0.5$ and neglected here. Much more important is the temperature dependence of

the liquid phase diffusivity and of the vapor pressure. The latter is described by the enthalpy of vaporization $\Delta H_{vap}$ (77.5 kJ mol$^{-1}$ for pure PEG-4), which characterizes the exponential decrease of vapor pressure with temperature.

In addition, for particles under dry and cold conditions the condensed phase diffusivity, $D_c(x_w, T)$, with its strong temperature dependence has to be considered. We describe this by an Arrhenius function

$$D_c(x_w, T) \propto e^{-E_{act}/(RT)}, \tag{7}$$

where $E_{act}$ is the diffusion activation energy. From our measurement (see Fig. A2) we estimate a mean diffusion activation energy $E_{act} \approx 300$ kJ mol$^{-1}$ for PEG-4 in sucrose for the measured RH range.

Figure 10 shows the evaporative loss of PEG-4 for a 100 nm radius particle at RH = 38 % for different altitudes (specified by the corresponding temperatures). Calculated equilibration times are compared with characteristic $1/e$ lifetimes of PAHs in the gas phase due to reaction with OH, see blue vertical band. OH-initiated reactions are the main reactive loss channel of 2- to

4-ring PAHs in the ambient atmosphere, resulting in calculated lifetimes of generally less than one day (Atkinson and Arey, 1994; Keyte et al., 2013). For each panel of Fig. 10, we make different assumptions regarding the temperature dependence of $D_c$.

In Fig. 10a, we first ignore the temperature dependence of the diffusion constant for $T < 10$ °C, i.e. we assume $D_c(x_w, T < 10$ °C$) = D_c(x_w, T = 10$ °C$)$. For all temperatures between 10 °C and -40 °C this yields diffusive equilibration times of less than

half an hour. Under these circumstances, the diffusive equilibration times are shorter than OH-induced loss times. Therefore, pollutant molecules diffusing through the organic matter of an aerosol particle in a similar manner as PEG-4 through sucrose at 10 °C, are not protected against chemical degradation, and long-range transport is not facilitated.

However, when we extrapolate the diffusion constant for T < °C with an Arrhenius relation consistent with the experimental data for sucrose (Fig. A1) corresponding to a diffusion activation energy $E_{act} = 300$ kJ mol$^{-1}$, the PEG-4 molecules (or other

pollutants it represents) are extremely well protected against evaporation. Figure 10b shows that for temperatures $T < -10$ °C diffusive equilibration times are longer than one week. A temperature of -10 °C corresponds to a seasonally averaged altitude of about 3 km in the mid-latitudes, and even lower in wintertime. Thus, a very efficient long-range transport mechanism of these pollutants protected by other organic molecules appears to be possible.

There are a number of caveats that must be discussed when applying this simple calculation to long-range transport of PAHs:

(I)   In the initial rapid upward transport a part of the volatile compound might be lost.

(II)   PEG-4 might be a bad proxy for PAH.

(III)   Sucrose might be a bad proxy for secondary organic matter.




(IV) $D_c = f(T)$ might not be Arrhenius-like, note that our extrapolation is based on measurements in a small temperature range.

(V) A protective cage effect might be undermined by

    a   rapid diffusion of the oxidant, i.e. the OH radical, in the condensed phase,

5         b   condensed phase photolysis of the PAHs.

Point (I) can be discarded. When an air mass is transported up, e.g. convectively, temperature decreases while relative humidity increases, which lets the vapor pressure of pollutant molecules decrease but keeps $D_c(x_w, T)$ high; hence, even more pollutant molecules can be taken up. When this humid air mass now mixes with the dry ambient air, RH drops (here to 38 %) and both water and PAH molecules start diffusing out of the particle, but water with the much higher vapor pressure equilibrates much

more quickly, sealing off the particle and trapping the PAH molecules.

Point (II) is likely irrelevant as well. PEG-4 and 3- to 4-ring PAHs in the ambient atmosphere have comparable masses (150-250 $\mathrm{g\,mol^{-1}}$) and the heats of vaporization (heats of sublimation for PAHs) are even higher for most PAHs than for PEG-4, providing a stronger T-dependence (Oja and Suuberg, 1998).

Point (III) is of real concern, as it is clear from our measurements of the diffusivity of water molecules in various organic

substances (Lienhard et al., 2015) that sucrose has particularly low water diffusivity and high $E_{act}$ compared to other organics. Assuming that diffusivities of organic species in various hosts scale like the diffusivity of water, from Table A1 for water diffusivity of Lienhard et al. (2015) we would need to reduce $E_{act} = 300\,\mathrm{kJ\,mol^{-1}}$ for PEG-4 in sucrose to $\sim 250\,\mathrm{kJ\,mol^{-1}}$ in levoglucosan, $\sim 225\,\mathrm{kJ\,mol^{-1}}$ in levoglucosan/$\mathrm{NH_4HSO_4}$, and $\sim 100\,\mathrm{kJ\,mol^{-1}}$ in $\alpha$-pinene based SOA, but increase to 325 $\mathrm{kJ\,mol^{-1}}$ in shikimic acid. Thus, diffusing molecules will typically have a lower $E_{act}$ in organic aerosols than in sucrose.

However, as the diffusion activation energy correlates with the heat of vaporization, a higher activation energy than that of water is expected for molecules such as PAH. These issues are addressed in Fig. 10c.

Point (IV) is of similar concern, as the limited temperature range of the measurements renders the extrapolation to lower temperatures using the Arrhenius equation questionable.

Point (V) is hard to judge. Concerning reaction with OH radicals, Davies and Wilson (2015) showed that reactions of citric

acid (CA) aerosols with OH are slowed when the aerosol assumes a viscous state. The reacto-diffusive length of OH is very small. Conversely, CA cannot rapidly diffuse from the core of the particle to the reactive shell, which establishes a core with the original CA concentration and an outer shell depleted in CA and enriched in degradation products. This is clear evidence that the diffusivity-induced protection effect works, sufficient to dismiss point (V)a. However, little is known about point (V)b. While photolysis plays a secondary role in the transformation of gaseous PAHs and the particle-associated processes are

usually slower than the gas-phase ones, direct or assisted photolysis plays nevertheless a relevant role in the transformation of particle-associated PAHs, given the longer atmospheric lifetimes of particulate PAHs (Vione et al., 2004).

In response to points (III) and (IV) above we performed another calculation of equilibration times with a reduced diffusion activation energy, $E_{act} = 200\,\mathrm{kJ\,mol^{-1}}$, which is shown in Fig. 10c. This might be more characteristic for atmospheric organic aerosols than the 300 $\mathrm{kJ\,mol^{-1}}$ for the PEG-4/sucrose system. Of course, the reduction in $E_{act}$ reduces the diffusive protection





against evaporation, but similar equilibration times as for $300 \, \mathrm{kJ \, mol^{-1}}$ are found at $10 \, ^\circ\mathrm{C}$ lower temperatures. Specifically, at $T = -20 \, ^\circ\mathrm{C}$, corresponding to typically $5 \, \mathrm{km}$ altitude in the mid-latitude troposphere, the equilibration time is $\tau \sim 11.5 \, \mathrm{d}$. This demonstrates that diffusion limitations can severely increase the equilibration times of atmospheric SOA particles and thereby enhances the long-range transport potential of trace pollutants and underlines the importance of accounting for the

5   temperature dependence of diffusivities. Clearly, to make more quantitative statements, a more realistic description using atmospheric trajectories would have to be considered.





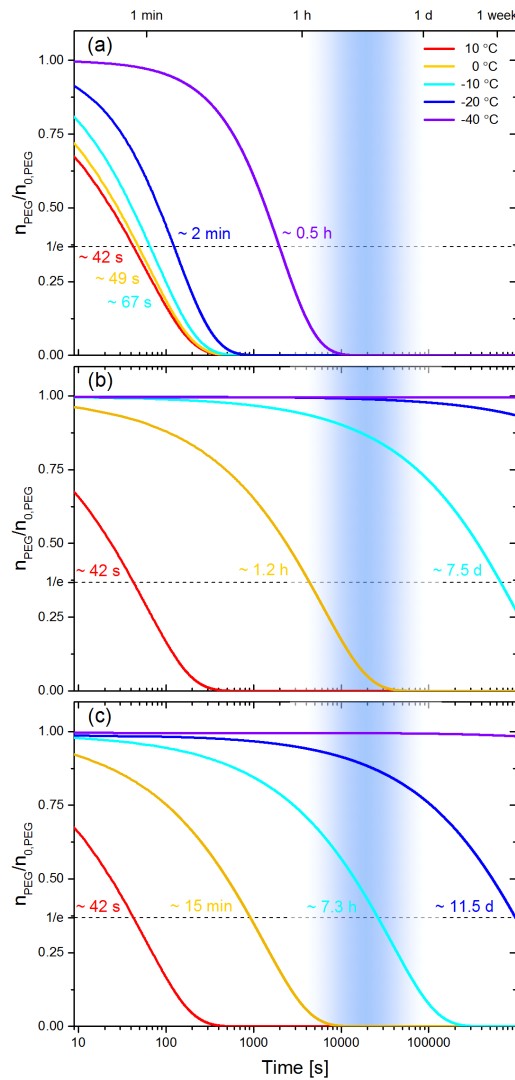

**Figure 10.** Equilibration times of 100-nm radius sucrose/PEG-4/water particles with initial $f_{PEG} = 0.04$ at RH = 38 % for different temperatures (as color-coded in upper right corner). (a) Ignoring the temperature dependence of the diffusion constant for $T < 10\,^{\circ}C$, i.e. assuming $D_c(x_w, T < 10\,^{\circ}C) = D_c(x_w, T = 10\,^{\circ}C)$. (b) Extrapolating the diffusion constant for $T < 10\,^{\circ}C$ with an Arrhenius relation consistent with the experimental data for sucrose (Fig. A2) corresponding to a diffusion activation energy $E_{act} = 300\,\mathrm{kJ\,mol}^{-1}$. (c) Like (b), but assuming $E_{act} = 200\,\mathrm{kJ\,mol}^{-1}$, believed to be more characteristic for atmospheric organic aerosols. For all calculations ambient pressure is chosen to match with temperatures in mid-latitudes, and the partial pressure of the solute (PEG-4) is zero. Labels indicate times after which the number of PEG-4 molecules $n_{PEG}$ within the particle is reduced to $1/e$ of the initial value $n_{0,PEG}$. Blue shaded region: Typical lifetimes of polycyclic aromatic hydrocarbons (PAHs) in the gas phase due to reaction with hydroxyl radicals (Atkinson and Arey, 1994; Keyte et al., 2013).





## 6    Conclusions

We have presented a method to quantitatively retrieve the diffusivity of volatile organic compounds in viscous matrices using a diffusion model and the evaporative radius change of a levitated single aerosol particle. Comparison with viscosity data of aqueous sucrose reveals a break-down of the Stokes-Einstein relation for PEG-4 at the high sucrose mole fraction range investigated in this study, albeit not as pronounced as for the diffusivity of water. This suggest that the kinetic gas to particle partitioning of volatile species of comparable or smaller $r_H$ is closer to equilibrium partitioning as expected from viscosity data of SOA. As diffusivity data in aqueous matrices at low water content is very sparse and is insufficient to reveal a clear dependence on the size of the diffusing molecule, more measurements are needed before definite conclusions can be drawn. In addition, the strong temperature dependence of our data and the simple model calculations for tropospheric conditions underline the importance of temperature-dependent diffusivity measurements to reliably assess the atmospheric impact of condensed-phase diffusion limitations.

## Appendix A

### A1    PEG-4 water activity

The water activity of PEG-4 was determined using a water activity meter (AquaLab water, Model 3B, Decagon Devices, USA) at 20 °C. In the diffusion model, the water activity dependence was parametrized as the cubic polynomial

$$x_w(a_w) = -0.07809 + 2.22455a_w - 1.85277a_w^2 + 0.70016a_w^3. \tag{A1}$$

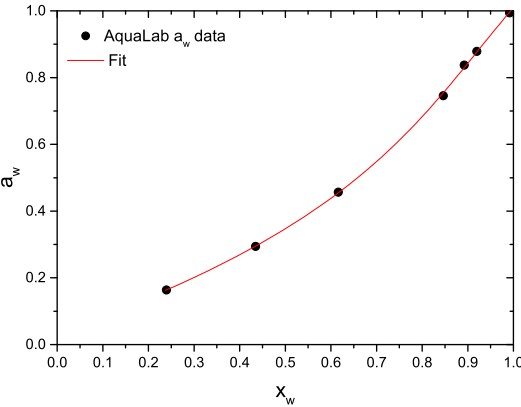

**Figure A1.** Water activity of PEG-4 (black) at 20 °C and parametrization used in diffusion model (red).





## A2 Density

To calculate partial molar volumes, the sucrose density data by Rosenbruch et al. (1975) and Fucaloro et al. (2007), the water density given by the Handbook of Chemistry and Physics (Haynes, 1996) and the partial molar volumes/density data of Cibulka (2016) and Klimaszewski et al. (2015) for PEG-4 were used. The partial molar volume of water $v_w$, sucrose $v_{suc}$ and PEG-4 $v_{\mathrm{PEG}}$ (in $\mathrm{cm^3\,mol^{-1}}$) were parametrized as functions of temperature $T$ (in K) and weight fractions of the individual components $w_{suc}$, $w_{\mathrm{PEG}}$ and $w_w$ using

$$v_w(T) = a_0 + a_1 T + a_2 T^2, \tag{A2}$$

with $a_0 = 26.40692$, $a_1 = -0.06068$ and $a_2 = 1.09701 \times 10^{-4}$,

$$v_{suc}(T, w_{suc}) = b_0 + b_1 w_{suc} + b_2 w_{suc}^2 + b_3 w_{suc} T + b_4 T, \tag{A3}$$

with $b_0 = 153.887$, $b_1 = 36.87558$, $b_2 = 6.22896$, $b_3 = -0.11518$ and $b_4 = 0.19375$, and

$$v_{\mathrm{PEG}}(T, w_{\mathrm{PEG}}) = c_0 + c_1 w_{\mathrm{PEG}} + c_2 w_{\mathrm{PEG}}^2 + c_3 T, \tag{A4}$$

with $c_0 = 123.73647$, $c_1 = -9.35554$, $c_2 = 16.79153$ and $c_3 = 0.14273$. The density $\rho(T)$ of the ternary system is calculated from the weight fractions, the partial molar volumes and the molar masses $M_{suc}$, $M_{\mathrm{PEG}}$ and $M_w$

$$\rho^{-1}(T) = \frac{w_{suc}}{M_{suc} v_{suc}} + \frac{w_{\mathrm{PEG}}}{M_{\mathrm{PEG}} v_{\mathrm{PEG}}} + \frac{w_w}{M_w v_w}. \tag{A5}$$

## A3 Wilson activity coefficient parametrization

The activity coefficients of a multi-component system can be parametrized in terms of binary system activity data using Wilson's equation

$$\ln \gamma_k = -\ln \left[ \sum_{j=1}^{N} x_j \Lambda_{kj} \right] + 1 - \sum_{i=1}^{N} \frac{x_i \Lambda_{ik}}{\sum_{j=1}^{N} x_j \Lambda_{ij}}, \tag{A6}$$

where $\Lambda_{ij}$ are interaction parameters with $\Lambda_{ii} = 1$, $\Lambda_{ij} > 0$ $\forall i,j$ (Orye and Prausnitz, 1965). In the case of the ternary sucrose/PEG-4/water-system the equation is given by six parameters $\Lambda_{sw}$, $\Lambda_{ws}$, $\Lambda_{pw}$, $\Lambda_{wp}$, $\Lambda_{sp}$ and $\Lambda_{ps}$ where $s$ is sucrose, $w$ is water and $p$ is PEG-4. The interaction parameters $\Lambda_{sw}$ and $\Lambda_{ws}$ were fitted to the binary Zobrist et al. (2011) water activity parametrization for sucrose. The interaction parameters $\Lambda_{pw}$ and $\Lambda_{wp}$ were fitted to the bulk water activity measurements of the binary PEG-4/water mixture (see Fig. A1, black circles). $\Lambda_{sp}$ and $\Lambda_{ps}$ were subsequently determined using the $\gamma_{\mathrm{PEG}}$ presented in Fig. 5 (a) (filled circles) and ternary system bulk water activity measurements. This procedure results in $\Lambda_{sw} = 0.15657$, $\Lambda_{ws} = 6.38724$, $\Lambda_{pw} = 0.25599$, $\Lambda_{wp} = 3.90419$, $\Lambda_{sp} = 1.81059$ and $\Lambda_{ps} = 7.64132 \times 10^{-16}$ and yields $\gamma_{\mathrm{PEG}}$ values that compare well with measured $\gamma_{\mathrm{PEG}}$ values (see Fig. 5b and a).




### A4 Diffusion activation energy

The measurement data was fitted to an Arrhenius temperature dependence and an exponential concentration dependence according to the function

$$D_{\mathrm{PEG}} = D_{\mathrm{PEG},0} \exp[-(E_{act}/R)(1/T - 1/T_0) - \beta(x_{suc} - x_{suc,0})]. \tag{A7}$$

5  We obtain an averaged value of $E_{act} = 295 \pm 27 \ \mathrm{kJ\,mol^{-1}}$, see Fig. A2. Due to the small concentration range in our measurements, this simple concentration dependence seems justified. However, in general, the diffusion activation energy $E_{act}$ is likely to be a function of $x_{suc}$.

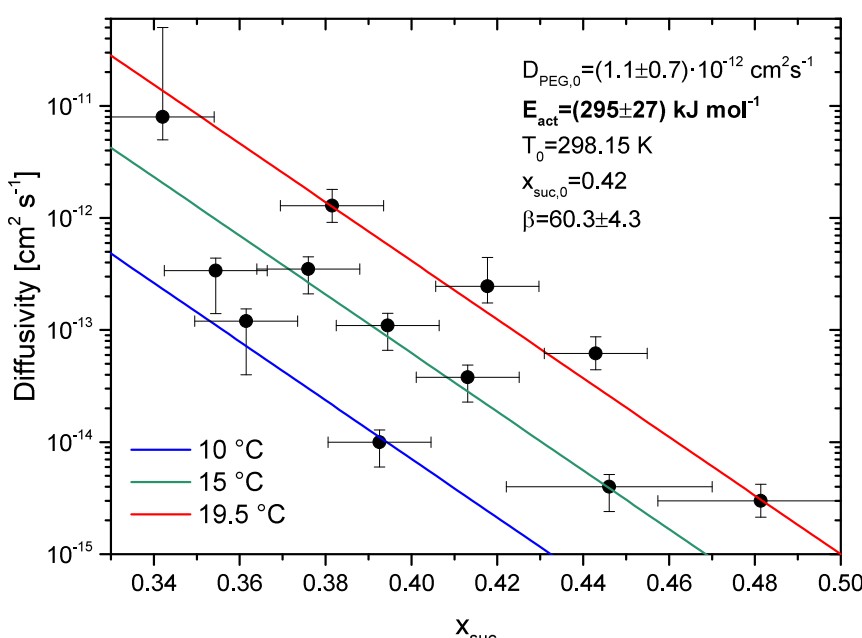

**Figure A2.** The activation energy $E_{act}$ is determined using a fit to the diffusivity data at 19.5 °C, 15 °C and 10 °C following Eq. (A7). $E_{act}$ is assumed to have no concentration dependence.



*Competing interests.* The authors declare that they have no conflict of interest.

*Acknowledgements.* The authors acknowledge funding by the Swiss National Science Foundation (grant No. 200020 146760/1).





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
