# Peer review of "Diffusivity measurements of volatile organics in levitated viscous aerosol particles"

_Atmospheric Chemistry and Physics, 2017_

## Referee Comment (RC1) · Anonymous Referee #1 · 6 Apr 2017

Review of "Diffusivity measurements of volatile organics in levitated viscous aerosol particles".

This manuscript focuses on the diffusion of volatile organics in highly viscous aerosol particles. This topic has come to the forefront of atmospheric research due to the recent qualitative field measurements that suggested diffusion rates in atmospheric particles is slow under some conditions and may be important for climate and air quality predictions. Until recently, measurements of diffusion coefficients of organics in particles have been missing. This paper provides some of the first quantitative measurements on this topic.

The scientific quality of manuscript is high. The conclusions are supported by the

data, and the manuscript represents a substantial contribution to scientific progress within the scope of Atmospheric Chemistry and Physics. The scientific results and conclusions are also presented in a clear and concise way. I recommend publication after the authors have had a chance to address the following very minor comments:

1) Page 14, Line 15-16. "it is evident from the sharp RH steps in Fig. 6 that there is no time delay between particle response and the model response". I suspect that there is a small time delay (on the order of tens of minutes) between particle response to RH and the model response, since previous research has shown that there is a kinetic limitation to water uptake in sucrose-water particles at low relative humidities. For clarity, I suggest the authors state the small time delay between particle response to RH and the model response (assuming there is one), and then argue that this small time delay is not important in the current experiments since this time delay is very short compared to the time needed for PEG to evaporate from the particles.

2) Page 9. The authors list changes implemented to adapt the model to the sucrose/PEG-4/water-system. In a couple of places it would be useful to indicate the accuracy of these changes. Specifically for points 3-4 it would be helpful to state roughly the accuracy of the methods used to calculate water content and density if known. In addition, for point 5, the authors state "Assuming zero PEG-4 concentration at infinite distance from the particle". Please state the accuracy of this assumption if known.
* * *

---

## Referee Comment (RC2) · Anonymous Referee #2 · 21 May 2017

The authors present an important addition to the literature on diffusivity in potentially viscous aerosol states. The focus on organic compounds takes the research in a direction that is essential to quantify any potential effects. I recommend publication after some minor general comments are addressed.

Is there a potential for a 'non zero' limiting partial pressure of the evaporating organic to bias any inferred diffusion coefficient in these experiments? I appreciate this might be very difficult to quantify, but please add to any appropriate references. One might expect this would only really be a problem for low viscosity states.

I am a little confused by the comment in point '2' on page 9 that you assume the condensed phase diffusion of PEG to not rely on its own concentration? Given the

mixture composition will dictate the changing viscosity, dosnt this restrict the use of your fit in other studies? Or, is it that you are targeting the use of your fit in other mixtures where you only need to track water?

How do you account for changing solubility as the droplet composition changes in additional studies? Given you account for non-ideality as noted in point 5, does this treatment persist between each condensed shell? It seems in section 4.1 that you optimize the activity coefficient, but how much error is inherent from other experimental sensitivities in this procedure? In other words, would it be possible to show the impact on inferred diffusion if you used a purely theoretical calculation of activity coefficients? It might be in more complex systems, unless you could use a robust multidimensional optimization strategy, this would be required anyway. I appreciate you discuss some of these issues in section 4.2 but the paper would benefit from a 'pure' sensitivity simulation. Figure 5 is very useful by the way.

Could you please add more specific details on how you optimize your activity coefficient values? Is this done by a specific algorithm, set tolerance or by eye?

On this note in section 4.2 the authors comment on the challenge in propagating errors in this model. With the availability of some algorithms to do this, it would be great to add a note on where others might be able to obtain the code to perform such calculations or how best to collaborate on this.

I find section 5 an excellent addition to this paper and studies restricted to studying a small subset of atmospherically relevant organics. It begs the question how we might move towards more general quantification. Perhaps a useful addition is to ensure we start to study affects in ensemble populations through chamber and box-model studies. The impact on size distribution should also be studied.
* * *

---

## Author Comment (AC1) · 14 Jun 2017

The authors would like to thank Referee #1 for the recommendation for publication and the helpful comments and questions. We address Referee #1's comments in our response given below and will incorporate the corresponding changes in a revised version of our manuscript.

Referee #2 : 1) Page 14, Line 15-16. "it is evident from the sharp RH steps in Fig. 6 that there is no time delay between particle response and the model response". I

suspect that there is a small time delay (on the order of tens of minutes) between particle response to RH and the model response, since previous research has shown that there is a kinetic limitation to water uptake in sucrose-water particles at low relative humidities. For clarity, I suggest the authors state the small time delay between particle response to RH and the model response (assuming there is one), and then argue that this small time delay is not important in the current experiments since this time delay is very short compared to the time needed for PEG to evaporate from the particles.

Authors' response: We will make the following changes to the manuscript: "It is evident from the sharp RH steps in Fig. 6 that there is little time delay between the particle response and the model response (blue curve). For the lowest RH, the response time is at the most $\sim 15$ min, which is very small compared to the timescales over which the diffusivity coefficients were determined ($> 10$ h). This validates the assumption that water diffusion is sufficiently fast and does not have to be treated explicitly under the experimental conditions considered in this study. "

Referee #1 : 2) Page 9.The authors list changes implemented to adapt the model to the sucrose/PEG-4/water-system. In a couple of places it would be useful to indicate the accuracy of these changes. Specifically for points 3-4 it would be helpful to state roughly the accuracy of the methods used to calculate water content and density if known. In addition, for point 5, the authors state "Assuming zero PEG-4 concentration at infinite distance from the particle". Please state the accuracy of this assumption if known.

Authors' response: The accuracy of point 3, which refers to the use of ZSR, cannot be tested throughout the whole concentration range. However, we performed several bulk water activity measurements for non-saturated solutions (with the accessible concentration range being $x_{suc} < 0.1$). For $f_{PEG} < 0.6$ the ZSR based calculated ternary solution molality of PEG-4 for a given $a_w$ deviated from the true solution values by less than 8 %. For $f_{PEG} < 0.2$, which is closer to our experimental range, the ZSR based calculated ternary solution molality of PEG-4 for a given $a_w$ deviated

from the true solution values by less than 3 %. In general, as we approach very small PEG-4 concentrations, we expect the accuracy of ZSR predictions to be given mainly by our knowledge of the molality of sucrose. Similarly (concerning point 4), a subset of the above mentioned non-saturated solutions were used to perform density measurements with a pycnometer that can be compared to the partial molar volumes approach. The room temperature pycnometer measurements agreed with partial molar volume predictions within 1 %.

We will add the following sentence to the description under point 3: "The accuracy of this estimation cannot be tested throughout the whole concentration range. However, we performed several bulk water activity measurements for non-saturated solutions (with the accessible concentration range being $x_{suc} < 0.1$). For $f_{PEG} < 0.6$ the ZSR based calculated ternary solution molality of PEG-4 for a given $a_w$ deviated from the true solution values by less than 8 %. For $f_{PEG} < 0.2$, which is closer to our experimental range, the ZSR based calculated ternary solution molality of PEG-4 for a given $a_w$ deviated from the true solution values by less than 3 %. In general, as we approach very small PEG-4 concentrations, we expect the accuracy of ZSR predictions to be given mainly by our knowledge of the molality of sucrose. "

We will also add to point 4: "For estimating the accuracy of this approach a subset of the non-saturated solutions discussed under 3. were used to perform density measurements with a pycnometer that can be compared to the partial molar volumes approach. The room temperature pycnometer measurements agreed with partial molar volume predictions within 1 % "

Point 5 was previously discussed in Huisman et al. (2013), section 2. Our flow rates can maintain a $p_\infty/p < 1$ % (Zhang and Davis, 1987). We will make the following change to the manuscript: "...where $\mathcal{D}_g$ is the gas phase diffusion constant of PEG-4, $x_n$ is the mole fraction of PEG-4 in shell $n$, $\gamma_{PEG}$ is the activity of PEG-4 in the ternary system (see Appendix A3), $p^0$ is the pure component vapor pressure of PEG-4 and $R$ is the

universal gas constant. The flow rates used in the experiments are sufficiently high to maintain the vapor pressure far from the particle, $p_\infty$, at less than 1 % of the vapor pressure above the particle, justifying the assumption (Zhang and Davis, 1987).

**References**

Huisman, A. J., Krieger, U. K., Zuend, A., Marcolli, C., and Peter, T.: Vapor pressures of substituted polycarboxylic acids are much lower than previously reported, Atmos. Chem. Phys., 13, 6647–6662, 10.5194/acp-13-6647-2013, 2013.
Zhang, S. and Davis, E. J.: Mass transfer from a single micro-droplet to a gas flowing at low reynolds number, Chem. Eng. Commun., 50, 51–67, 10.1080/00986448708911815, http://dx.doi.org/10.1080/00986448708911815, 1987.

---

## Author Comment (AC2) · 14 Jun 2017

The authors would like to thank Referee #2 for the recommendation for publication and the helpful comments and questions. We address Referee #2's comments in our response given below and will incorporate the corresponding changes in a revised version of our manuscript.

Referee #2 : Is there a potential for a 'non zero' limiting partial pressure of the evaporating organic to bias any inferred diffusion coefficient in these experiments? I appreciate this might be very difficult to quantify, but please add to any appropriate

references. One might expect this would only really be a problem for low viscosity states.

Authors' response: Due to the finite experimental observation time of the particle within the EDB, there will, strictly speaking, always be a very small amount of PEG-4 left within the particle at the end of the experiment, thereby causing a non-zero PEG-4 partial pressure. If the "leftover" concentration of PEG-4 at the end of an experiment was substantial, this could in principle bias the total particle size inferred from the shifts of the LED Mie resonances for previous times. However, for the following reasons we deem a potential bias of our inferred diffusion coefficients caused by a substantial non-zero partial pressure of PEG-4 at the end of our experiments unlikely:

- The particles were equilibrated in the non condensed-phase diffusion limited regime until no shrinkage due to evaporation was observable. Given a $24$ h observation period, our limit of sensitivity to radius change corresponds to a partial pressure of $\sim 3.6 \times 10^{-7}$ Pa (Huisman et al., 2013). Under the assumption of ideality, this corresponds to $x_{PEG} \sim 4 \times 10^{-5}$ in a homogeneous particle at $19.5$ °C. Thus, we do not expect a large error in total radius due to a residual non-zero partial pressure at the end of an experiment.

- $f_{PEG}$, i.e. PEG-4 content at the beginning of the experiment calculated from the Mie resonance shifts and the end size is consistent with the $f_{PEG}$ of the solutions that were prepared to generate the particles.

- The retrieval of the PEG-4 condensed phase diffusivities relies on the rate of radius change and is not primarily dependent on the total radius.

Referee #2 : I am a little confused by the comment in point '2' on page 9 that you assume the condensed phase diffusion of PEG to not rely on its own concentration?

Given the mixture composition will dictate the changing viscosity, doesn't this restrict the use of your fit in other studies? Or, is it that you are targeting the use of your fit in other mixtures where you only need to track water?

Authors' response: The diffusivity retrieval in this study relies on the simplifying assumption, i.e. Fickian diffusion (the diffusivity is independent of concentration of PEG-4), that can be made in the limit of the low concentrations of PEG-4 that were used. This approach will indeed break down for higher PEG-4 concentrations, which inherently necessitate a much more complicated treatment, i.e. solving the nonlinear diffusion equation. Hence, instead of covering the whole concentration range, this study aims at providing diffusivity measurements for a PEG-4 like molecule in the viscosity range we investigated. To clarify which viscosity range was addressed, we will make the following changes to the manuscript: "Bosse ideal mixing yields good results for aqueous sucrose and suggests that, if applicable to our ternary system, sucrose provides the dominant contribution to the mixture viscosity due to its very high "pure component" viscosity and the contribution of PEG-4 can be treated as water in first approximation. Based on this approach, our 19.5 C data corresponds to a viscosity range of $10^5 - 10^8$ Pa s."

Referee #2: How do you account for changing solubility as the droplet composition changes in additional studies? Given your account for non-ideality as noted in point 5, does this treatment persist between each condensed shell?

Authors' response: As we are studying a Fickian diffusion process, which is driven by concentration gradients alone, during which our system neither phase separates nor partially effloresces, solubility is not an issue. The activity coefficient only influences the effective PEG-4 vapor pressure and therefore our model just requires treatment of the activity coefficient for the outermost shell. Consequently, in its current form, our model is not applicable to systems that undergo phase changes.

Referee #2: It seems in section 4.1 that you optimize the activity coefficient, but how much error is inherent from other experimental sensitivities in this procedure? In other words, would it be possible to show the impact on inferred diffusion if you used a purely theoretical calculation of activity coefficients? It might be in more complex systems, unless you could use a robust multidimensional optimization strategy, this would be required anyway. I appreciate you discuss some of these issues in section 4.2 but the paper would benefit from a 'pure' sensitivity simulation.

Authors' response: In the non condensed-phase diffusion limited regime, which was used to determine the activity coefficients, the accuracy with which activity coefficients can be determined hinges on our knowledge of the pure component vapor pressure of PEG-4, which is better than $\pm 10$ % (95 % confidence interval) (Krieger et al., 2017). As pointed out by Referee #2, the sensitivity of the inferred diffusivities on activity coefficient was addressed in the y-error-bars given in Fig. 7 of section 4.2, for which the flux into the gas phase was multiplied or divided by a factor of 3. Below $D_{PEG} < 10^{-13}$ cm$^2$ s$^{-1}$, this treatment had no influence on the retrieved diffusivities and the error bar given in the figure reflects a conservative estimate that the error in manually fitting the slope is not bigger than a factor of 1.4. Above $D_{PEG} \sim 10^{-13}$ cm$^2$ s$^{-1}$, the error bars given in the figure are dominated by the multiplication/division of the flux into the gas phase by a factor of three. While we did not explicitly calculate the activity coefficients in each time step using theoretical calculations, the factor three that we applied widely exceeds the range given by AIOMFAC (see Fig. 5). In the condensed phase diffusion limited regime, the diffusivities are not very sensitive to the activity coefficient. We chose the RH range of our measurements conservatively in order to avoid strong sensitivity of diffusivity to activity coefficients. In this sense, only very small activity coefficients (of less than$\sim$ 0.1) are likely to bias inferred diffusivities. However, the activity coefficient measurements and AIOMFAC calculations both indicate values close to ideality and > 1.

Referee #2: Could you please add more specific details on how you optimize your activity coefficient values? Is this done by a specific algorithm, set tolerance or by eye

Authors' response: The activity coefficients were optimized by eye. To address Referee # 2 comment, we will add the following sentences to the revised version of the manuscript: "We determine $\gamma_{PEG}$ at high RH by freely adjusting its value in the diffusion model to reproduce the experimental radius curve presented in Fig. 4a (red) while $D_{PEG}$ was set to a constant value of $1.0 \times 10^{-9}$ cm$^2$ s$^{-1}$. The best activity coefficient fit was determined by eye. The chosen diffusivity is well within the gas phase diffusion limited regime ..."

Referee #2: On this note in section 4.2 the authors comment on the challenge in propagating errors in this model. With the availability of some algorithms to do this, it would be great to add a note on where others might be able to obtain the code to perform such calculations or how best to collaborate on this.

Authors' response: We will add the following sentence to the manuscript:"As the determination of $D_{PEG}$ still requires knowledge of $\gamma_{PEG}$, we rely on Wilson's approach for multi-component systems (Orye and Prausnitz (1965)), Appendix A3) to parametrize $\gamma_{PEG}$ based on binary and ternary bulk data as well as EDB $\gamma_{PEG}$ measurements in the gas diffusion limited regime. The code of our diffusion model is available upon request."

Referee #2: [on section 5] It begs the question how we might move towards more general quantification. Perhaps a useful addition is to ensure we start to study

affects in ensemble populations through chamber and box-model studies. The impact on size distribution should also be studied

Authors' response: We agree with the reviewer, but these suggestions are beyond the scope of the present paper.

**References**

Huisman, A. J., Krieger, U. K., Zuend, A., Marcolli, C., and Peter, T.: Vapor pressures of sub-stituted polycarboxylic acids are much lower than previously reported, Atmos. Chem. Phys., 13, 6647–6662, 10.5194/acp-13-6647-2013, 2013.

Krieger, U. K., Siegrist, F., Marcolli, C., Emanuelsson, E. U., Gobel, F. M., Bilde, M., Marsh, A., Reid, J. P., Riipinen, I., Myllys, N., Hyttinene, N., Kurtén, T., Bannan, T. and Topping, D.: to be submitted to, Atmos. Meas. Tech., 2017.